# Iridium-Catalyzed and pH-Dependent Reductions of Nitroalkenes to Ketones

**DOI:** 10.3390/molecules27227822

**Published:** 2022-11-13

**Authors:** Tingting Wang, Changmeng Liu, Dong Xu, Jiaxi Xu, Zhanhui Yang

**Affiliations:** Department of Organic Chemistry, College of Chemistry, Beijing University of Chemical Technology, Beijing 100029, China

**Keywords:** iridium catalysis, nitroalkenes, ketones, acidic water

## Abstract

A highly chemoselective conversion of *α,β*-disubstituted nitroalkenes to ketones is developed. An acid-compatible iridium catalyst serves as the key to the conversion. At a 2500 *S/C* ratio, nitroalkenes were readily converted to ketones in up to 72% isolated yields. A new mechanistic mode involving the reduction of nitroalkene to nitrosoalkene and *N*-alkenyl hydroxylamine is proposed. This conversion is ready to amplify to a gram-scale synthesis. The pH value plays an indispensable role in controlling the chemoselectivity.

## 1. Introduction

Conjugated nitroalkenes are important synthetic precursors that can be converted to a variety of useful compounds, such as nitroalkanes, oximes, ketones, hydroxylamines, amines, and others [1,2,3,4,5,6]. Compared with ketone formation from nitroalkanes [7], the direct transformation of nitroalkenes to ketones is a step-economic and important synthetic tool, which has served as an essential step in the manufacture of amphetamine and methylamphetamine [8,9]. Previous methods used to realize this transformation mainly relied on the use of excessive metals or metal salts as reducing reagents (Figure 1A, i) [10,11,12,13,14,15,16,17,18,19]. This traditional strategy mainly suffers from the generation of large amounts of hazardous metal salt waste (30 equivalents at most) and low chemoselectivity in some cases. Metal hydrides, such as lithium selectride and tributyltin hydride, are also applicable (stoichiometric amount), but anhydrous conditions are required for the reduction step (Figure 1A, ii) [20,21]. Other methods also include electrochemical reduction with lead electrodes (Figure 1A, iii) [22] and biocatalytic reduction with enzymes (very low yield) [23,24]. In the transition metal catalyzed hydrogenation of nitroalkenes to nitroalkanes, ketones were serendipitously observed as byproducts [25,26]. From the perspective of green chemistry, it is necessary to develop a simple, mild, and efficient method to reducibly transform nitroalkenes to ketones, although such a transformation is very challenging empirically due to the higher reactivity of the C=C bond than the nitro group.

In the above reports, ketones were smoothly formed under acidic reducing conditions. Under basic or neutral conditions, low yields of ketones and large amounts of by-products were generated. Thus, we believe that a catalysis strategy under acidic conditions will provide a good solution to the challenge. In the core of this strategy lies the catalyst, which must be acid compatible. The acidic conditions should not prevent, or ideally promote, the catalyst evolution in the catalytic cycle. Since 2017, we have designed and synthesized a series of iridium complexes with 2-(4,5-dihydroimidazol-2-yl)pyridines as ligands, and developed their applications in the functional group transformations in a green catalysis manner under acidic conditions [27,28,29,30,31,32,33,34,35,36,37,38], including the pH-dependent reduction of nitroalkenes to nitroalkanes (Figure 1B) [33]. These catalysts have shown their strong capability in a series of green syntheses with formic acid as the reducing reagent. Iridium hydrides were identified as the key reductive species to transfer the hydrides to electrophilic species. The relative stability of iridium hydrides under acidic conditions holds the response for the occurrence of these reductions. The added acid promotes the catalytic cycle by activating the starting materials [28,29,30,31,32,33,34,35] and directs the reaction intermediates to different evolutions.

Based on the understanding of the mechanism of our acid-tolerable iridium catalysis, we decided to tackle the nitroalkene-to-ketone challenge by using our iridium catalysis strategy. To our delight, a very low catalyst loading in combination with formic acid as the reducing regent and mineral acid as the acidic promoter readily converted *α,β*-disubstituted nitroalkenes to ketones in water with a good chemoselectivity and reasonable yields (Figure 1C). The present reduction is chemoselectively orthogonal to our previous reduction under neutral conditions that converted the same type of nitroalkenes to nitroalkanes (Figure 1B) [33]. Under acidic conditions (pH = 1.6), the nitro group is predominantly reduced (Figure 1C), while under neutral conditions (pH = 7.1), the C=C bond is exclusively reduced [33]. Our iridium catalysis strategy enables a green nitroalkene-to-ketone transformation, without the employment of harsh conditions or the generation of stoichiometric transition-metal waste, and mechanistically provides a new reduction mode for ketone synthesis from nitroalkenes. Compared with previous work (Figure 1A), our strategically novel method is advantageous as it has mild conditions, simple experimental operations, is user friendly, and does not generate stoichiometric transition-metal waste.

## 2. Results and Discussion

Our previous work on the reduction of nitroalkenes to nitroalkanes demonstrated that a **C3** catalyst with a 4-methoxy on the pyridinyl ring showed the highest activity, and it was also able to catalyze the generation of oxime and methyl ketone from 1-phenyl-2-nitroprop-1-enes [33]. Thus, this catalyst was selected in the reaction condition optimization (Table 1) to probe other reaction parameters. 1-Phenyl-2-nitroprop-1-ene (**1a**) was used as the model substrate. Upon workup, the crude reaction mixtures were directly submitted to ^1^H NMR analysis, and the distributions of the starting material and all products were easily obtained (for the calculation method, see the Appendix A). Under acidic conditions, three products were obtained, that is, nitroalkane (**2a**), oxime (**3a**), and the desired methyl ketone (**4a**). The oximes can be regarded as masked ketones. It is well documented that oximes are able to completely hydrolyze to ketones upon treatment with hydrochloric acid [11]. Thus, the most important goal of the optimization is to prevent the generation of nitroalkane **2a** to the largest extent. At first, we optimized the amount of formic acid (entries 1–3) at 5000 S/C (substrate/catalyst molar ratio). With 4, 8, or 16 equivalents of formic acid, nitroalkene **1a** was almost completely consumed (< 1% residual), and the yield of nitroalkane **2a** remained at the same levels (12–14%). However, the larger the amount of formic acid used, the larger the amount of ketone **4a** observed. This phenomenon was ascribed to the more efficient hydrolysis of oxime **3a** to **4a** under more acidic conditions, as proved by the decreasing yield of oxime **3a** against ketone **4a** (entries 1–3). To further decrease the ratio of nitroalkane **2a** and to facilitate the hydrolysis of oxime **3a** to ketone **4a**, we subsequently tried adding sulfuric acid to the reaction mixture (entries 4–9). The sulfate anion (SO_4_^2−^) is noncoordinating, and its presence did not affect the catalyst activity. Contrastingly, the addition of a high volume of coordinating chloride anion would poison the catalyst. That is why we did not add hydrochloric acid. Satisfyingly, the addition of 25 or 50 μL of concentrated sulfuric acid completely inhibited the formation of nitroalkane **2a** and oxime **3a**. However, ketone **4a** was only generated in 10–33% yields, depending on the amount of added sulfuric acid (pH values). The larger the amount of sulfuric acid added, the lower the yield of ketone **4a** and the lower the conversion of nitroalkene **1a**. It should be clarified that the inhibition of the reduction was not caused by the sulfate anion (SO_4_^2−^), but by the high concentration by proton (H^+^), which greatly hindered the dissociation of formic acid and subsequent iridium hydride formation (vide post and see the mechanistic discussion section). The addition of ethanol partially alleviated the inhibition effect (entry 6), so ethanol was added throughout the next optimization. Increasing the amount of catalyst allowed the complete conversion of the starting material, and the yield of nitroalkane was suppressed to < 7% level (entries 7–9). When using dilute sulfuric acid (3.68 mol/L) instead of concentrated acid, the yield of **4a** increased to 72% (entry 10) while the yields of oxime and nitroalkane were 25% and 3%, respectively. One of the principles that chemists adhere to in the catalysis field is to use as little catalyst as possible to achieve results that are as good as possible. Thus, we turned back to the original catalyst loading (S/C = 5000). To obtain as high of a yield of ketone as possible, we decided to treat the reaction mixture after reduction. To our delight, reduction at a 5000 or 2500 S/C ratio followed by stirring with HCl for 15, 30, or 60 min delivered ketone **4a** in a 91%, 94% or 96% yield, respectively (entries 11–13). Other catalysts (**C1–C2** and **C4–C9**) (Table 1, entries 14–21), which were prepared in our previous publications [27,28,29,30,31,32,33], were further screened. No oxime was produced under these conditions. The catalysts bearing 4-dimethylamino- and 4-diethylamino- at each pyridine ring showed excellent efficiency, delivering **4a** in 89% and 87% yields, respectively, with the full conversion of **1a** (entries 14 and 15). In contrast, the remaining catalysts resulted in lower yields (3–55%) of **4a**, and the starting material **1a** remained incompletely consumed (entries 16–21). Generally, the catalysts bearing electron-donating groups were superior to those bearing electron-withdrawing groups. This substituent effect was also observed in our previous studies [28]. According to the above results, it is obvious that **C3** was the most robust catalyst, and we therefore obtained the optimal conditions (entries 13).

The substrate scope was further explored (Table 2). The reductions of not only (2-nitroprop-1-en-1-yl)benzene (**1a**), but of its 4-methoxy (**1b**), 4-methylthio (**1c**), 4-methyl (**1d**), 4-fluoro (**1e**), 4-chloro (**1f**), and 4-bromo (**1g**) relatives all gave desired products in reasonable to good (34–72%) yields (Table 2, entries 1–7). Other viable substrates included those bearing electron-withdrawing groups on the aromatic ring. For example, 4-trifluoromethyl (**1h**), 4-cyano (**1i**), and 4-nitro- (**1j**) and 4-methanesulfonly-substituted (**1k**) nitroalkenes were converted to the corresponding ketones in 32–65% yields (Table 2, entries 8-11). The nitro groups residing on aryl and alkenyl showed different reactivity (Table 2, entry 10). The former was quite resistant to the reducing conditions. Although nitrile functionality was prone to hydrolysis under acidic conditions, it survived well during the reduction of **1i** (Table 2, entry 9). The subjection of 1-phenyl-2-nitrobut-1-ene (**1l**), a substrate bearing a longer substituent (ethyl) in place of methyl, to the standard conditions gave 1-phenylbutan-2-one (**4l**) in a 34% yield (Table 2, entry 12). To our surprise, the reduction of indol-3-yl nitroalkene **1m** failed, with the starting material completely recovered (Table 2, entry 13). Nitroalkenes of other substitution patterns, for example, **1n**, **1o**, and **1p**, were reduced to the corresponding nitroalkenes in excellent yields (Table 2, entry 14), as reported in our previous work [33]. These examples demonstrated the necessary role of an alkyl substituent on the nitro-adjacent alkenyl carbons.

In previous reductions of nitroalkenes to ketones [10,11,23,24], nitroalkanes were demonstrated or proposed as the first-order reduction products, which were further reduced to nitrosoalkanes (Figure 2a). The fast tautomerization of nitrosoalkanes produced oximes, which collapsed to ketones (vide post). In our reductions, nitroalkanes were also observed as minor products in most cases. Therefore, we wondered whether our reduction would follow the same mechanism as that depicted in Figure 2a. We first submitted nitroalkane **2a** to our optimal conditions (Figure 2b, i). To our surprise, neither oxime **3a** nor ketone **4a** was observed at all. Instead, nitroalkane **2a** was recovered quantitatively, demonstrating its invulnerability to the iridium-catalyzed reduction. We also demonstrated that nitroalkane **2a** was quite stable under the acidic reaction and workup conditions in the absence of a catalyst (Figure 2b, ii), as no Nef reactions occurred [7]. These observations clearly confirmed that nitroalkanes were not precursors to ketones in our reductions, and a different mechanism is around the corner.

A plausible mechanism for the reduction of nitroalkenes to ketones was proposed (Figure 3). The ionization of formic acid (Figure 3a) and iridium chloride (**C3**) (Figure 3b, step i) delivers the formate anion and iridium cation (**A**), respectively. They combine together to form iridium formate (**B**) (step ii). Factors such as having too high of a concentration of the proton and chloride anion that affect these two steps also affect the formation of iridium formate (**B**), and finally affect the final catalytic reductions. The β-elimination of **B** gives rise to iridium hydride (**C**), which serves as the catalyst resting state and transient reducing reagent (step iii). On the other hand, under acidic conditions, nitroalkenes (**1**) are activated by the protonation of the nitro group (**5**). Possibly, it is the protonation that renders the nitro moiety more reactive than the alkenyl moiety when exposed to iridium hydride (**C**). As a consequence, the hydroiridation occurs across the N=O bond at the nitro moiety to generate intermediates **D** (step iv). In comparison, in our previous work under neutral conditions, the hydroiridation between the same iridium hydride and the same nitroalkenes takes place across the C=C bond to finally give nitroalkane products [33]. In the presence of formic acid, **D** evolves to *N*-alkenyl-*N*-hydroxyl hydroxylammoniums (**6**) (step v), which collapse to protonated nitrosoalkenes (**7**) via dehydration [39,40,41,42,43], accomplished by the restart of the catalytic cycle (the regeneration of **A**). The protonated nitrosoalkenes (**7**), more reactive than protonated nitroalkenes (**1**) [39,40,41,42,43], receive the hydrides from iridium hydrides **C** and are further reduced to *N*-alkenyl hydroxylamines (**8**) (Figure 3c). Since the suggested nitrosoalkenes (**7**) and *N*-alkenyl hydroxylamines (**8**) are active intermediates for the reduction, they cannot be observed or isolated. The tautomerization of **8** affords oxime **3**. There exist two viable routes to ketones **4** from oximes **3**. The direct hydrolysis of **3** under acidic conditions gives **4** [11]. Alternatively, **3** is further catalytically reduced to imines **10**, which is spontaneously hydrolyzed to **4** [23]. Such a reduction sequence for the generation of ketones from nitroalkenes is quite different from the previously reported sequence that involves reductions of nitroalkene to nitroalkane and nitrosoalkane (Figure 2a). It must be mentioned that the formation of hydrogen was observed in all cases, and this can be mechanistically elucidated by the reaction between protons and iridium hydrides (Figure 3b).

To demonstrate the synthetic applications of our method, we performed a gram-scale synthesis of **4a** (Figure 4). This ketone was further isolated by column chromatography in a 70% yield as a yellowish oil. Numerous methods have been developed from this ketone to synthesize central nervous system stimulants such as (±)-, *R*-, or *S*-amphetamine [44,45,46,47,48,49], (±)- or *S*-methylamphetamine [50,51,52], and (±)-ethylamphetamine [53]. Amphetamine is a prescribed medication for treating attention-deficit hyperactivity disorder and narcolepsy. Ethylamphetamine (trade name Apetinil or Adiparthrol) was used as an anorectic or appetite suppressant. The syntheses of these bioactive compounds from **4a** imply the importance of our reduction for preparing arylmethyl ketones.

## 3. Materials and Methods

### 3.1. Materials and Instruments

Unless otherwise noted, all reagents and solvents were used directly as commercially received. Column chromatography was performed using silica gel (normal phase, 200−300 mesh) from a Branch of Qingdao Haiyang Chemical, with petroleum ether (PE, 60−90 °C fraction) and ethyl acetate (EA) as the eluent. Reactions were monitored by thin-layer chromatography on GF254 silica gel plates (0.2 mm) from the Institute of Yantai Chemical Industry. The plates were visualized under UV light. ^1^H NMR, ^13^C NMR, and ^19^F NMR spectra were measured with a Bruker 400 spectrometer in CDCl_3_ with tetramethylsilane (TMS) as an internal standard.

Nitroalkenes (**1a**–**m**) were previously synthesized in our work [33] according to reported procedures [54,55,56,57]. The catalysts **C1**–**C9** and their solutions in water were prepared according to our previous work [33].

### 3.2. General Procedure for Reduction of Nitroalkenes to Ketones

To a 25 mL round-bottom flask, nitroalkenes **1** (1 mmol), EtOH (3 mL for **1a,b,d,e,h,l**; 4 mL for **1c,f,g,i,j,k,m**), the solution of catalyst **C3** (2 mL, 0.0002 mol/L for S/C = 2500; 0.005 mol/L for S/C = 100 in deionized water), formic acid (320 μL, 8 equiv., 8 mmol), and sulfuric acid (3.7 mol/L, 100 μL) were sequentially added. The mixture was stirred at 80 °C for 3 h; then, HCl (4 mL) was added and stirring was continued for another 1 h. After cooling to room temperature, diluting with water (8 mL), and extracting with ethyl acetate (8 mL × 3), the organic phase was washed with saturated sodium bicarbonate and dried over anhydrous sodium sulfate. The organic phase was evaporated under reduced pressure and the crude product was purified by flash column chromatography to give **4a** to **4l**.

#### 3.2.1. *1-Phenylpropan-2-one* (**4a**) 

CAS No. 103-79-7 [58]. Yellow oil, 96 mg, yield 72%, R*_f_* = 0.27 (PE/EA (*v/v*) = 10:1). ^1^H NMR (400 MHz, CDCl_3_) δ 7.38–7.15 (m, 5H), 3.68 (s, 2H), 2.14 (s, 3H). ^13^C NMR (101 MHz, CDCl_3_) δ 206.4, 134.3, 129.5, 128.8, 127.1, 51.1, 29.3.

#### 3.2.2. *1-(4-Methoxyphenyl)propan-2-one* (**4b**)

CAS No. 122-84-9 [59]. Yellow oil, 72 mg, yield 44%, R*_f_* = 0.15 (PE/EA (*v/v*) = 10:1). ^1^H NMR (400 MHz, CDCl_3_) δ 7.11 (d, *J* = 8.6 Hz, 2H), 6.87 (d, *J* = 8.6 Hz, 2H), 3.79 (s, 3H), 3.62 (s, 2H), 2.13 (s, 3H). ^13^C NMR (101 MHz, CDCl_3_) δ 206.9, 158.8, 130.5, 126.4, 114.3, 55.3, 50.2, 29.2.

#### 3.2.3. *1-(4-(Methylthio)phenyl)propan-2-one* (**4c**) 

CAS No. 88356-92-7 [60]. Yellow oil, 61 mg, yield 34%, R*_f_* = 0.24 (PE/EA (*v/v*) = 10:1). ^1^H NMR (400 MHz, CDCl_3_) δ 7.23 (d, *J* = 8.3 Hz, 2H), 7.12 (d, *J* = 8.4 Hz, 2H), 3.65 (s, 2H), 2.48 (s, 3H), 2.15 (s, 3H). ^13^C NMR (101 MHz, CDCl_3_) δ 206.4, 137.4, 131.2, 130.0, 127.2, 50.5, 29.4, 16.1.

#### 3.2.4. *1-(p-Tolyl)propan-2-one* (**4d**) 

CAS No. 2096-86-8 [61]. Yellow oil, 83 mg, yield 56%, R*_f_* = 0.34 (PE/EA (*v/v*) = 10:1). ^1^H NMR (400 MHz, CDCl_3_) δ 7.14 (d, *J* = 7.8 Hz, 2H), 7.08 (d, *J* = 8.1 Hz, 2H), 3.64 (s, 2H), 2.33 (s, 3H), 2.13 (s, 3H). ^13^C NMR (101 MHz, CDCl_3_) δ 206.7, 136.8, 131.3, 129.5, 129.3, 50.7, 29.2, 21.2.

#### 3.2.5. *1-(4-Fluorophenyl)propan-2-one* (**4e**) 

CAS No. 459-03-0 [62]. Yellow oil, 78 mg, yield 52%, R*_f_* = 0.19 (PE/EA (*v/v*) = 10:1). ^1^H NMR (400 MHz, CDCl_3_) δ 7.12–7.03 (m, 2H), 6.97–6.90 (m, 2H), 3.59 (s, 2H), 2.08 (s, 3H). ^13^C NMR (101 MHz, CDCl_3_) δ 206.1, 162.1 (d, *J_C-F_
*= 246.4 Hz), 131.04 (d, *J_C-F_
*= 8.1 Hz), 130.0 (d, *J_C-F_
*= 3.0 Hz), 115.7 (d, *J_C-F_
*= 21.6 Hz), 50.0, 29.4. ^19^F NMR (377 MHz, CDCl_3_) δ -115.8.

#### 3.2.6. *1-(4-Chlorophenyl)propan-2-one* (**4f**) 

CAS No. 5586-88-9 [62]. Yellow oil, 91 mg, yield 54%, R*_f_* = 0.19 (PE/EA (*v/v*) = 10:1). ^1^H NMR (400 MHz, CDCl_3_) δ 7.30 (d, *J* = 8.4 Hz, 2H), 7.12 (d, *J* = 8.4 Hz, 2H), 3.67 (s, 2H), 2.16 (s, 3H). ^13^C NMR (101 MHz, CDCl_3_) δ 205.7, 133.1, 132.7, 130.9, 128.9, 50.1, 29.5.

#### 3.2.7. *1-(4-Bromophenyl)propan-2-one* (**4g**) 

CAS No. 6186-22-7 [62]. Yellow oil, 87 mg, yield 41%, R*_f_* = 0.14 (PE/EA (*v/v*) = 10:1). ^1^H NMR (400 MHz, CDCl_3_) δ 7.46 (d, *J* = 8.4 Hz, 2H), 7.07 (d, *J* = 8.4 Hz, 2H), 3.66 (s, 2H), 2.16 (s, 3H). ^13^C NMR (101 MHz, CDCl_3_) δ 205.6, 133.2, 131.9, 131.3, 121.3, 50.3, 29.6.

#### 3.2.8. *1-(4-(Trifluoromethyl)phenyl)propan-2-one* (**4h**) 

(CAS No. 713-45-1) [62]. White solid, m.p. 30–32 °C, 118 mg, yield 58%, R*_f_* = 0.15 (PE/EA (*v/v*) = 10:1). ^1^H NMR (400 MHz, CDCl_3_) δ 7.58 (d, *J* = 8.0 Hz, 2H), 7.31 (d, *J* = 8.0 Hz, 2H), 3.78 (s, 2H), 2.19 (s, 3H). ^13^C NMR (101 MHz, CDCl_3_) δ 205.1, 138.3 (q, *J*_C-F_ = 2.0 Hz), 130.0, 129.4 (q, *J*_C-F_ = 32.3 Hz), 125.7 (q, *J*_C-F_ = 4.0 Hz), 124.3 (q, *J*_C-F_ = 273.0 Hz), 50.4, 29.7. ^19^F NMR (376 MHz, CDCl_3_) δ -62.6.

#### 3.2.9. *4-(2-Oxopropyl)benzonitrile* (**4i**) 

CAS No. 58949-75-0 [63]. White solid, m.p. 66–68 °C, 103 mg, yield 65%, R*_f_* = 0.12 (PE/EA (*v/v*) = 5:1). ^1^H NMR (400 MHz, CDCl_3_) δ 7.54 (d, *J* = 8.3 Hz, 2H), 7.23 (d, *J* = 8.2 Hz, 2H), 3.73 (s, 2H), 2.14 (s, 3H). ^13^C NMR (101 MHz, CDCl_3_) δ 204.4, 139.5, 132.4, 130.4, 118.7, 111.0, 50.4, 29.9.

#### 3.2.10. *1-(4-Nitrophenyl)propan-2-one* (**4J**) 

CAS No. 5332-96-7 [64]. Yellow oil, 71 mg, yield 40%, R*_f_* = 0.11 (PE/EA (*v/v*) = 5:1). ^1^H NMR (400 MHz, CDCl_3_) δ 8.20 (d, *J* = 8.7 Hz, 2H), 7.37 (d, *J* = 8.7 Hz, 2H), 3.85 (s, 2H), 2.25 (s, 3H). ^13^C NMR (101 MHz, CDCl_3_) δ 204.2, 147.3, 141.6, 130.6, 123.9, 50.2, 30.0.

#### 3.2.11. *1-(4-(Methylsulfonyl)phenyl)propan-2-one* (**4k**) 

CAS No. 88356-97-2 [65,66]. White solid, m.p. 78–80 °C, 68 mg, yield 32%, R*_f_* = 0.11 (PE/EA (*v/v*) = 2:1). ^1^H NMR (400 MHz, CDCl_3_) δ 7.91 (d, *J* = 8.3 Hz, 2H), 7.40 (d, *J* = 8.3 Hz, 2H), 3.84 (s, 2H), 3.06 (s, 3H), 2.24 (s, 3H). ^13^C NMR (101 MHz, CDCl_3_) δ 204.5, 140.5, 127.8, 50.3, 44.7, 30.0. HRMS (ESI): *m*/*z* calculated for [M+Na]^+^: 235.0399; found: 235.0396.

#### 3.2.12. *1-phenylbutan-2-one* (**4l**) 

CAS No. 1007-32-5 [67]. Colorless oil, 50 mg, yield 34%, R*_f_* = 0.45 (PE/EA (*v/v*) = 10:1). ^1^H NMR (400 MHz, CDCl_3_) δ 7.37–7.17 (m, 5H), 3.68 (s, 2H), 2.47 (q, *J* = 7.3 Hz, 2H), 1.02 (t, *J* = 7.3 Hz, 3H). ^13^C NMR (101 MHz, CDCl_3_) δ 209.0, 134.5, 129.4, 128.7, 126.9, 49.8, 35.2, 7.8.

## 4. Conclusions

In summary, we have realized the highly chemoselective conversion of α,β-disubstituted nitroalkenes to ketones by means of iridium catalysis in acidic water. At 2500 S/C ratio, a number of diversely substituted nitroalkenes were readily converted to ketones in 32–72% yields. The advantages of this method include the employment of midconditions, user- and environment-friendly operations, and no generation of stoichiometric transition-metal waste. The application of an acid-compatible iridium catalyst is key to the successful conversion. The chemoselectivity is governed by the pH value of the reaction media. The proposed mechanism involves the acid-controlled selective reduction of nitroalkenes to nitrosoalkenes and *N*-alkenyl hydroxylamines, providing a new mechanistic mode for nitroalkene-to-ketone transformations. The synthetic application of this conversion is demonstrated by a gram-scale synthesis. We hope our study will promote the green and easy preparation of amphetamine-type medications in medicinal chemistry.

## Data Availability

The data presented in this study are available on request from the corresponding author.

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
