# Peer review of "Iridium-Catalyzed and pH-Dependent Reductions of Nitroalkenes to Ketones"

_molecules, 2022, doi:10.3390/molecules27227822_

Round 1

Reviewer 1 Report

This manuscript describes the transformation of nitroalkenes to ketones catalyzed by iridium catalysis enables.

Chemoselective conversion of α,β-disubstituted nitroalkenes to ketones was achieved by means of iridium catalysis in the presence of formic acid in acidic water. It is noted that a reduction sequence for the generation of ketones from nitroalkenes is quite different from the previously reported sequence that involves reductions of nitroalkene to nitroalkane.

The reaction conditions was well optimized, and mechanistic study was well done,

Further, gram-scale synthesis was performed leading to bioactive compounds.

From there reasons, I conclude this work is suitable for publication to Molecules.

Before publication, please check the following small points.

(1) Scheme 3: It is better to show a counter anion Cl- in dot box of Ir complex.

(2) line 181: compound 4 is to bold style.

(3) line 129-138: About substrates scope, only methyl substituted alkene to form methyl ketones were focusing. How about the reactivity for other substituent in place of methyl group because tin hydride reduction  could apply styrene derivatives (ref 21).  If author has information about other types of substrates like longer chain substituents or hydrogen, it is better to show about the limitation.

Author Response

This manuscript describes the transformation of nitroalkenes to ketones catalyzed by iridium catalysis enables.

Chemoselective conversion of α,β-disubstituted nitroalkenes to ketones was achieved by means of iridium catalysis in the presence of formic acid in acidic water. It is noted that a reduction sequence for the generation of ketones from nitroalkenes is quite different from the previously reported sequence that involves reductions of nitroalkene to nitroalkane.

The reaction conditions was well optimized, and mechanistic study was well done,

Further, gram-scale synthesis was performed leading to bioactive compounds.

From there reasons, I conclude this work is suitable for publication to Molecules.

Before publication, please check the following small points.

(1) Scheme 3: It is better to show a counter anion Cl- in dot box of Ir complex.

Response: Added.

(2) line 181: compound 4 is to bold style.

Response: Style changed.

(3) line 129-138: About substrates scope, only methyl substituted alkene to form methyl ketones were focusing. How about the reactivity for other substituent in place of methyl group because tin hydride reduction  could apply styrene derivatives (ref 21).  If author has information about other types of substrates like longer chain substituents or hydrogen, it is better to show about the limitation.

Response: More substrate information are given. In the revised manuscript, we expanded the scope in Table 2, added:

“Subjection of 1-phenyl-2-nitrobut-1-ene (1i), a substrate bearing a longer substituent (ethyl) in place of methyl, to the standard conditions gave 1-phenylbutan-2-one (4l) in 34% yield. To our surprise, the reduction of indol-3-yl nitroalkene 1m failed, with the starting material completely recovered. Nitroalkenes of other substitution patterns, for example, 1n, 1o, and 1p, were reduced to the corresponding nitroalkenes in excellent yields, as reported in our previous work [33]. These examples demonstrated the necessary role of an alkyl substituent on the nitro-adjacent alkenyl carbons.”

Reviewer 2 Report

The manuscript titled Iridium-Catalyzed and pH-Dependent Reductions of Nitroalkenes to Ketones shows promising transformation. Some of the key features about the protocol: Excellent catalytic method, gives direct access to 2-arylacetone, development of a novel metal hydride catalytic system (possibly), gram scale conversion. line 33, 34, 35: english corrections, and a lot of other typos, overall english writing needs attention, please follow strict scientific writings.   following are some of the areas needs to be addressed for publication:
1) limited substrate scope, more substrate scope required possibly with different substitution patterns.   2) is the molar concentration of the sulfuric acid (3.68 M) adjusted before adding to the reaction. Then after addition, please mention the final molarity.   3) Using Iridium catalyst for the transformation is a good advancement however it required a lot more advantages clearly presented in introduction and summary compared to the previous method.

Author Response

The manuscript titled Iridium-Catalyzed and pH-Dependent Reductions of Nitroalkenes to Ketones shows promising transformation. Some of the key features about the protocol: Excellent catalytic method, gives direct access to 2-arylacetone, development of a novel metal hydride catalytic system (possibly), gram scale conversion.

line 33, 34, 35: english corrections, and a lot of other typos, overall english writing needs attention, please follow strict scientific writings.

Response: Corrected. The whole English writings have been carefully scrutinized, and the errors have been corrected..

following are some of the areas needs to be addressed for publication:

1) limited substrate scope, more substrate scope required possibly with different substitution patterns.

Response: More substrate information are given. In the revised manuscript, we expanded the scope in Table 2, and added:

“Subjection of 1-phenyl-2-nitrobut-1-ene (1i), a substrate bearing a longer substituent (ethyl) in place of methyl, to the standard conditions gave 1-phenylbutan-2-one (4l) in 34% yield. To our surprise, the reduction of indol-3-yl nitroalkene 1m failed, with the starting material completely recovered. Nitroalkenes of other substitution patterns, for example, 1n, 1o, and 1p, were reduced to the corresponding nitroalkenes in excellent yields, as reported in our previous work [33]. These examples demonstrated the necessary role of an alkyl substituent on the nitro-adjacent alkenyl carbons.”

2) is the molar concentration of the sulfuric acid (3.68 M) adjusted before adding to the reaction. Then after addition, please mention the final molarity.  

Response: Mentioned.

3) Using Iridium catalyst for the transformation is a good advancement however it required a lot more advantages clearly presented in introduction and summary compared to the previous method.

Response: Clearly presented. In the introduction section, we added “Compared with previous work (Scheme 1A), our strategically novel method is advantageous of mild conditions, simple experimental operations, user-friendliness, and no generation of stoichiometric transition-metal waste.” In the conclusion section, we add: “Advantages of this method include employment of mid conditions, user- and environment-friendly operations, and no generation of stoichiometric transition-metal waste.”

Round 2

Reviewer 2 Report

excellent work, the manuscript has been revised extensively with lot more information added.